# Institutional Violence Perpetrated Against Transgender Individuals in Brazilian Healthcare Services: A Conceptual Analysis Based on Foucault’s Microphysics of Power

**DOI:** 10.3390/ijerph22111655

**Published:** 2025-10-31

**Authors:** Gilberto da Cruz Leal, José Nildo de Barros Silva-Júnior, Quézia Rosa Ferreira, Thomas Oliveira Silva, Lícia Kellen de Almeida Andrade, Ana Luíza Brasileiro Nato Marques Assumpção, Mônica Cristina Ribeiro Alexandre d’Auria de Lima, Jaqueline Garcia de Almeida Ballestero, Inês Fronteira, Pedro Fredemir Palha

**Affiliations:** 1Ribeirão Preto College of Nursing, University of São Paulo, Ribeirão Preto 14040-902, Brazil; jose.nildo@usp.br (J.N.d.B.S.-J.); quezia@usp.br (Q.R.F.); liciaandrade@usp.br (L.K.d.A.A.); analuizaassumpcao@usp.br (A.L.B.N.M.A.); mcraadl@gmail.com (M.C.R.A.d.d.L.); jaqueline.almeida@usp.br (J.G.d.A.B.); palha@eerp.usp.br (P.F.P.); 2Public Health Research Centre, NOVA National School of Public Health, NOVA University Lisbon, 1600-407 Lisbon, Portugal; ines.fronteira@ensp.unl.pt; 3Comprehensive Health Research Center, NOVA National School of Public Health, NOVA University Lisbon, 1600-407 Lisbon, Portugal; 4Institute of Scientific and Technological Communication and Information in Health, Oswaldo Cruz Foundation, Rio de Janeiro 21040-900, Brazil; 5School of Public Health, University of São Paulo, São Paulo 01246-904, Brazil; thomasoliveirasilva@usp.br

**Keywords:** transgender persons, violence, health services, health services for transgender persons, gender-based violence

## Abstract

Institutional violence against transgender individuals in healthcare is a structural phenomenon with multifactorial roots and is embedded in social, cultural, political, and economic dynamics. Drawing on Michel Foucault’s concept of the microphysics of power, this study analyzes how such violence unfolds within Brazil’s healthcare system. For Foucault, power is not solely exercised through centralized institutions but is diffuse and present in everyday practices, relationships, and discourses. Institutional violence, while often manifesting in individual acts, is defined by its persistence and systematic occurrence over time. In healthcare settings, this violence assumes symbolic, structural, psychological, and physical forms ranging from neglect and verbal abuse to sexual violence. Foucault’s framework allows for a deeper understanding of how power relations perpetuate exclusionary and dehumanizing practices. By interpreting these dynamics through the Microphysics of Power, the study reveals that institutional violence against transgender individuals extends beyond explicit acts, encompassing routine interactions that reproduce inequality and restrict access to fundamental rights such as healthcare. These practices sustain a logic of control and exclusion that operates subtly but effectively within healthcare systems, reinforcing the marginalization of trans people and undermining their right to dignified and equitable care.

## 1. Introduction

Institutional violence perpetrated against transgender (trans) individuals in healthcare services is a structural phenomenon with multifactorial and multidisciplinary dimensions, permeating social, cultural, political, and economic spheres worldwide. These characteristics both reflect and perpetuate the inequalities faced by this population across various settings [1].

Leal et al. [2] reported that institutional violence against the trans population in healthcare services occurs when agents of public or private organizations—responsible for meeting users’ needs—fail in their duties, whether through explicit harmful attitudes and practices or through subtle actions that reinforce heterocisnormative norms.

The concept of heterocisnormativity, frequently referenced in this study, describes the social regulation and production of identities, genders, and bodies [3]. Its manifestation reveals how heterosexual norms, practices, bodies, and discourses are socially constructed and legitimized as sovereign, natural, and even compulsory. This can be observed from birth, when individuals are assigned colors, toys, and other items that perpetuate binary associations between boys and girls, demonstrating a powerful imposition linking biological sex and gender identity [4,5].

These global patterns of heteronormativity are also observable in Brazil, where the healthcare system mirrors both the advances and the persistent limitations in meeting the needs of trans people [6,7]. Costa et al. [6] noted that healthcare for the trans population in Brazil has been transitioning from a model focused exclusively on gender-affirming surgeries (such as transgenitalization and mastectomy) to one that is comprehensive and person-centered, particularly in major urban centers. However, Lima et al. [7] warned that healthcare for trans individuals in Brazil remains exclusionary, fragmented, and overly specialized, guided by curative actions and resembling care models that predate the establishment of the Unified Health System (Sistema Único de Sáude—SUS).

The purpose of this study is not to conduct an in-depth analysis of this issue, as such practices depend on the intentionality between users and healthcare services, as well as on market-driven dynamics. They also rely on other intermediate-level factors, such as public policy frameworks, professional training, and the influence of social movements, all of which shape the provision of healthcare for the trans population [7].

Despite advances in the legal and social recognition of gender identity in Brazil, such as the right to use one’s chosen name [8] and access to trans-specific healthcare [9], including the transsexualization process offered by SUS [10], healthcare institutions can still become spaces of discrimination, neglect, and dehumanization. This reveals a gap between the idealized policy framework and the lived experiences of trans individuals [2].

As a result of this contradictory scenario, the universal access of trans individuals to the constitutional right to health [11] remains compromised. This situation increases the social vulnerability of this population and contributes to increased morbidity, mortality, and psychological distress [12]. Therefore, there is an urgent need to analyze the power dynamics that sustain violent practices, moving beyond superficial explanations to uncover the deeper mechanisms operating within these institutions.

Michel Foucault’s concept of the microphysics of power [13] provides a valuable theoretical lens through which to understand the violence experienced by the trans population in healthcare settings. In Foucauldian terms, power does not reside solely within institutions or centralized authorities (such as governments or formal organizations); rather, it is diffused throughout everyday relationships, professional (care) practices, and the discourses of healthcare workers [13].

These practices and discourses are reflected in the behavior of healthcare workers, who, while influenced by both micro- and macrocontexts, also exert influence upon them, thereby reinforcing what is known as organizational culture [14]. This framework helps elucidate how heterocisnormative norms and values are reproduced in interactions between healthcare professionals and transgender individuals, transforming healthcare settings into arenas where bodies are disciplined and differences are controlled or excluded.

The health of the transgender population remains an issue that demands continued investigation and monitoring, particularly at a time when their rights are increasingly challenged and obstructed by political leaders of major world powers, such as the United States. In Brazil, as in many other countries, the political and social climate has become fertile ground for the spread of conservative, prejudiced, and discriminatory ideas and practices [15], especially since the late 2010s.

The scarcity of studies addressing violence against transgender individuals in healthcare services—especially regarding its manifestations and underlying motivations—combined with the ineffectiveness of public health and social protection policies aimed at this group and the inherent vulnerabilities of the population itself underscore the urgent need for comprehensive and effective monitoring of this phenomenon.

In light of the above, this study aims to examine the institutional violence perpetrated against transgender individuals in Brazilian healthcare services.

## 2. Methods

This is a conceptual analysis study grounded in Michel Foucault’s theoretical-analytical framework of the Microphysics of Power [13]. The analysis is supported by evidence synthesized in a previous systematic review of qualitative studies that documented the violent behaviors and attitudes of healthcare workers toward trans individuals in healthcare services [2].

These behaviors and attitudes were organized into seven thematic domains in the aforementioned review: denial of care; resistance to the use of chosen names and gender pronouns; barriers to accessing healthcare services; discrimination and stigma; professional insensitivity; lack of specialized care; and inadequate professional training within a binary-oriented system. However, the underlying mechanisms and rationalities that sustain these phenomena have not been explored in depth [2].

Building upon these findings, the present study adopts a Foucauldian conceptual approach to integrate and reinterpret the evidence identified in the systematic review [2]. The aim is to reveal patterns and dimensions of institutional violence, articulating them within a robust theoretical framework capable of elucidating the subtle and diffuse ways in which power operates within healthcare practices and care relationships.

Although the systematic review underpinning this study included evidence produced in diverse international contexts, this conceptual analysis was intentionally limited to the Brazilian context. This choice is justified by the Foucauldian understanding that relations of power and resistance are inherently local, manifesting in unique ways according to the historical, political, and structural specificities of each society [13]. Therefore, the analysis is restricted to Brazil, with the goal of deepening the understanding of how institutional violence is produced and reproduced within the Brazilian public healthcare system while also contributing to the international debate through a situated and critical reading of the power dynamics that structure public health systems.

Furthermore, the scarcity of studies addressing all levels of the healthcare network (Rede de Atenção à Saúde—RAS) in Brazil was a key factor motivating a broader and more comprehensive analysis of this scenario. Nonetheless, the present study placed particular emphasis on a detailed examination of primary healthcare (PHC), given its central role in coordinating and organizing care within the RAS.

The defining attributes of PHC, especially its function as the main entry point to healthcare services, the establishment of longitudinal relationships, the comprehensiveness of care, and its principles of regionalization, hierarchy, support systems, logistics, and governance, underscore its relevance in this context [16].

## 3. Violence: From a Broader Context to Its Institutionalization in Healthcare Services

First and foremost, it is essential to define the concept of violence to understand its dynamics and complexity. The literature characterizes violence as a polysemic and multifaceted phenomenon [17]. Given that this study is situated within the healthcare context, we adopt the definition proposed by the World Health Organization (WHO) [18], which defines violence as “the intentional use of physical force or power, threatened or actual, against oneself, another person, a group, or a community, that either results in or has a high likelihood of resulting in injury, death, psychological harm, developmental impairment, or deprivation.”

According to Meyran [17], violence is, in essence, socially interpreted—that is, what is or is not recognized as violence depends on specific sociohistorical and cultural contexts. Within this framework, violence against transgender people is often trivialized and normalized across multiple spheres of society [2].

Although violence is fundamentally a social issue and not exclusively a matter of concern for the healthcare sector, Minayo [19] emphasized that it becomes closely associated with health because of its direct impact on quality of life—through the physical, psychological, and moral harm it inflicts—and because it requires intervention and care within health services.

Institutional violence, therefore, is understood to emerge initially through individualized interactions between subjects within institutions. Institutionalization is defined as the recurrence and systemic diffusion of such episodes over time [20]. In the healthcare sector, specifically, this form of violence may manifest in symbolic, structural, psychological, and/or physical forms, expressed through neglect, verbal abuse, or even sexual assault [21].

## 4. Symbolic Violence

The concept of symbolic violence, developed by Pierre Bourdieu [22], refers to the imposition of meanings and cultural norms that are perpetuated through relations of power. It may manifest through language (for instance, the use of incorrect or inappropriate expressions and terms), cultural stereotypes (via the reinforcement of negative representations of certain groups), and coercive forms of education (by attempting to mold others according to a dominant and normative culture).

According to Borgert et al. [23], transgender individuals are frequently subjected to invasive questioning or to the denial of their identity. The authors argued that unwarranted curiosity by healthcare professionals about the personal lives of transgender people—at the expense of addressing their actual health concerns—can compromise the quality of care. Similarly, Goldenberg et al. [24] reported that conducting an initial clinical interview (anamnesis) that is unconditionally centered on gender identity, rather than on the patient’s health condition, constitutes a form of disrespect and thus represents one of the expressions of violence.

These practices exemplify how symbolic violence operates by conveying implicit messages of identity invalidation. Such experiences can lead transgender individuals to avoid healthcare services for fear of discrimination or humiliation.

In this sense, symbolic violence does not operate in isolation but rather interlaces with other forms of violence, reinforcing them within healthcare institutions. The repetition of everyday gestures of misrecognition and linguistic domination consolidates institutional norms that marginalize transgender individuals. As these microlevel practices become normalized, they uphold and legitimize broader patterns of exclusion and control, which Foucault described as the diffuse operation of power that disciplines bodies and regulates subjectivities [13].

Thus, symbolic violence becomes both a mechanism and an effect of institutional violence, revealing how power circulates through ordinary interactions and contributes to the systematic production of inequality.

## 5. Structural Violence

The concept of structural violence, developed by Johan Galtung [25], describes forms of violence embedded within social, economic, and political structures. This type of violence emerges when institutional systems perpetuate inequality by depriving certain groups of access to essential resources, rights, and opportunities.

According to Galtung [25], structural violence is characterized by systemic inequality (through the exclusion of minorities from spaces of power and decision-making), invisibility (as it is often normalized and therefore goes unnoticed), and cumulative impact (because of the physical, psychological, and social harm it inflicts over time).

Trindade [26] demonstrated that violence against transgender individuals is expressed through the denial of the right to gender-affirming procedures, the lack of qualified or willing professionals, and the prejudiced attitudes of workers across various service sectors. These attitudes delegitimize the work of healthcare teams involved in providing care. Similarly, Silva et al. [27] reported that most transgender individuals do not feel treated in a humanized manner within healthcare services, resulting in their needs often remaining unmet.

With respect to access to healthcare services, Costa et al. [6] reported that among the 626 Brazilian transgender individuals surveyed, 43% had avoided seeking care simply because they identified as trans. Moreover, 58% began avoiding healthcare services after experiencing discrimination. A U.S.-based study involving 91 transgender individuals [28] reported that 28% stopped seeking healthcare services following discriminatory incidents. These findings underscore that institutional violence is not a localized or isolated phenomenon but rather a widespread issue across diverse contexts.

From this perspective, structural violence functions as the macroframework within which symbolic and interpersonal forms of violence are reproduced. Institutional arrangements, bureaucratic routines, and normative expectations sustaining healthcare systems serve to naturalize exclusionary practices and conceal their violent effects [13]. When viewed through a Foucauldian lens, such structures operate as disciplinary mechanisms that shape behavior, regulate access to care, and produce hierarchies of legitimacy among subjects.

Consequently, structural violence is both sustained and sustained by everyday microviolences, creating a cyclical dynamic that perpetuates institutional violence as a normalized expression of power within healthcare environments.

## 6. Psychological Violence

Psychological violence manifests through actions, behaviors, or omissions that undermine the dignity, subjectivity, and emotional well-being of transgender individuals. It encompasses the denial of chosen names and pronouns, negligence or refusal of care, blaming the trans person, bullying, harassment, and pathologization of transgender identities, among other forms [23]. This type of violence may be expressed through verbal communication, including threats, devaluation, or verbal abuse, and through nonverbal behaviors, such as isolation, disparaging facial expressions or glances, emotional neglect, and inappropriate tone of voice.

A study conducted in Turkey [29] revealed violent attitudes among healthcare workers toward transgender individuals. Statements such as “If I had known you were trans, I wouldn’t have touched you” were reported during medical encounters, clearly illustrating explicit expressions of repulsion and rejection.

Leal et al. [2] documented cases in which transgender individuals were subjected to compulsory psychological or psychiatric treatment by healthcare providers, authorized by family members but without the consent of the individuals themselves. This scenario highlights not only the symbolic and institutional invisibility of trans people but also their lack of autonomy over their own bodies and decisions. Although such situations represent a form of psychological violence, they may also be understood as physical violence, insofar as they involve the regulation of the body through the imposition of medical procedures.

According to Pedra et al. [30], when individuals who do not conform to heterosnormative standards are marginalized, the dominance of the so-called “normal” subject is reinforced. In this context, recurrent experiences of violence against the Lesbian, Gay, Bisexual, Transgender, Intersex, Queer/Questioning, Asexual/Aromantic, Pansexual, Nonbinary, and other (LGBTQIAPN+) populations lead individuals to question and constrain their existence in society, generating insecurity, vulnerability, and emotional distress, which often manifests as sadness and anxiety [31].

From the perspective of Foucault’s microphysics of power, such practices exemplify how power in healthcare environments is not centralized solely within management structures or public policies but rather circulates through everyday interactions, professional routines, and institutional norms [13]. Addressing a transgender person by their birth name instead of their chosen name, for instance, is not merely an administrative error; it is a microphysical manifestation of power that reinforces binary gender norms and produces psychological suffering [2].

Similarly, when healthcare professionals assume the authority to assess whether a patient is “truly” trans, they position themselves as regulators of identity, exercising power over the legitimacy of a person’s existence. These seemingly minor and routine acts reveal how power operates within the microspaces of clinical practice, validating certain bodies and identities while delegitimizing others [13].

In this sense, psychological violence against transgender individuals is not only interpersonal but also institutional, embedded in the subtle mechanisms of control and normalization that characterize modern healthcare systems.

## 7. Physical Violence

Physical violence manifests through actions that cause bodily harm, pain, or physical suffering and is often associated with negligence or abuse of power. In the study by Kosenko et al. [32], participants reported insensitive and dangerous procedures performed by healthcare workers, such as performing surgical interventions without the use of anesthetic drugs. Such situations pose a serious threat to the lives of transgender individuals, who, as discussed later, already face reduced life expectancy compared with the general population.

Performing medical procedures without the individual’s consent, handling instruments roughly, or conducting invasive examinations without justification also constitute forms of physical violence against transgender individuals.

According to Söest and Bryant [33], violence manifests at multiple levels within healthcare systems. However, these forms of violence are interrelated and mutually reinforcing, ultimately culminating in institutional violence. Thus, violence assumes specific characteristics and expressions depending on the institutional mechanisms, interpersonal interactions, and power dynamics operating within the healthcare context.

In this context, physical violence functions as the most visible expression of a continuum of power relations that permeate healthcare environments. The neglect of bodily integrity both reflects and reinforces the symbolic and structural dimensions of violence, revealing how disciplinary practices materialize in the regulation, correction, and control of trans bodies [34].

From a Foucauldian perspective, such practices exemplify the transformation of institutions into spaces of normalization, where violence ceases to appear as an exception and becomes inscribed within the very structure of care. In this sense, physical violence both exposes and consolidates the institutional mechanisms through which bodies are rendered docile and compliant with normative expectations.

## 8. The History of Violence Against Transgender People in Brazil

According to Larrat [35] and Garcia et al. [15], the social position historically assigned to the LGBTQIAPN+ population in Brazil—particularly transgender people—is the outcome of a long historical process rooted in the colonial period. Since that time, the Church has played a central role in disseminating the Western Christian worldview, which has associated gender identities and sexual orientations with moral transgression.

For centuries, these identities were regarded as sins, pathologies, or even crimes. A significant milestone in the global process of depathologizing trans identities was the 11th edition of the International Classification of Diseases (ICD-11). In this new version, the term “transsexuality” was removed from the chapter “Personality and Behavioral Disorders,” subchapter “Gender Identity Disorders,” and relocated to a new chapter entitled “Conditions Related to Sexual Health.” Since then, it has been reclassified as “gender incongruence” [36].

Although dissident genders and identities are no longer classified as diseases or considered crimes in Brazil, the Church continues to portray these individuals as sinful and promiscuous [37]. This worldview—rooted in a cisheteronormative, patriarchal, and machista structure consolidated over centuries—had a particularly profound impact during the military dictatorship, when the militarization of security reinforced mechanisms of social control. Today, it continues to perpetuate a system of exclusion and marginalization. As a result, prejudice and discrimination not only persist but are also renewed and reinforced across multiple social spheres, obstructing access to fundamental rights and perpetuating structural inequalities that deeply affect the lives of LGBTQIAPN+ individuals [35].

This historical legacy of control and exclusion continues to shape contemporary institutional norms. The colonial and authoritarian logics that once sought to discipline bodies through religion, medicine, and the State have been incorporated into the bureaucratic, legal, and clinical routines that now regulate who is recognized and how [37]. In healthcare settings, for example, these legacies manifest through standardized procedures, moral judgments, and administrative practices that reproduce cisnormative assumptions, legitimizing certain identities while rendering others invisible [34].

Thus, the institutional apparatus functions as a modern expression of historical mechanisms of normalization and subjugation, sustaining the continuity of violence in subtle yet pervasive ways.

## 9. Challenges Faced by the Trans Population in Health Services

For Judith Butler [38], most people live in harmony with the gender socially assigned to them. However, some individuals suffer precisely because they are compelled to conform to dominant social norms—as is the case for transgender people—a condition that inevitably denies the deepest sense of who they are or aspire to be.

According to Leal et al. [2], one of the primary challenges faced by the Brazilian transgender population concerns access to healthcare services, or rather, the lack thereof. The scarcity of trained and qualified professionals, coupled with discriminatory and stereotyped attitudes and institutional prejudice, often drives transgender individuals away from seeking medical care.

The widespread practice of categorizing individuals strictly according to biological or anatomical criteria (i.e., male or female sex) also represents a serious barrier for transgender people. This approach disregards the diversity of gender identities and perpetuates a reductionist and exclusionary framework. By relying exclusively on such criteria, gender identity—as an essential dimension of subjectivity and human dignity—is overlooked [39].

The challenges faced by this population within healthcare settings are numerous. Monteiro and Brigeiro [40] and Gomes et al. [41] reported that transgender individuals frequently encounter hostile situations, including the refusal of care by healthcare professionals, administrative constraints that lead to the omission of physical examinations, difficulties in understanding medical guidance or prescriptions, and the nonacceptance of social names or self-referenced pronouns by healthcare staff.

Kcomt [42] further reported that transgender people experience exceptionally high rates of discrimination and face more barriers to care than other groups do—conditions that severely compromise healthcare access and illustrate the pervasive nature of institutional mistreatment. These findings indicate that the patterns discussed here are not isolated or anecdotal but are widely documented across different contexts.

Moreover, factors such as race, class, and sexuality intersect to intensify and complicate the forms of violence experienced by transgender people. Ghosh [43] highlights that the absence of an intersectional approach in transgender health research represents a significant gap in contemporary scientific production, as it limits understanding of the multiple layers of exclusion that shape these experiences. The failure to consider how racism, classism, and homophobia intertwine with transphobia contributes to the normalization of inequalities and the reproduction of social hierarchies within healthcare institutions themselves.

Thus, understanding institutional violence against transgender people requires an intersectional perspective—one that reveals how different axes of power converge in producing vulnerabilities and reinforcing the norms that sustain institutional cisheteronormativity.

## 10. The Microphysics of Power and Institutional Violence

According to Foucault [13], diffuse and relational power is exercised at the individual level and within social contexts, permeating various aspects of everyday life. Foucault examines how power relations are constructed and maintained at both the macroscopic level (within institutions of power such as government) and the microscopic level (in interactions between individuals and smaller institutions, such as family and healthcare services, among others).

Interactions between service users and healthcare workers, for example, can reflect the dynamics of the Microphysics of Power. Healthcare workers, by virtue of their specialized knowledge and authority, may assume an authoritarian position regarding treatment decisions and access to healthcare. These interactions tend to be marked by unequal power relations, in which patients feel unable to question or negotiate their care options [20]. Furthermore, these micropower spaces create situations where healthcare professionals tend to disregard the will of service users, such as the use of social names merely to exert power over the other.

Similarly, Santos and Pereira [44] highlighted that the transgender population, when seeking care at SUS, frequently encounters bureaucratic demands that hinder access to services ranging from outdated systems that do not recognize social names to the imposition of unsolicited procedures on the basis of the pathologization of their identities. Healthcare workers, in exercising microdecisions in the daily provision of care, become agents of an institutional normativity that reinforces exclusions. Frequent delays in service, refusal to address specific needs, differential treatment, or omission of necessary information for continuity of care are concrete expressions of this diffuse power, which legitimizes itself within the routine of the service.

In his work Discipline and Punish [34], Foucault introduces the concept of the “docile body”, which refers to a body shaped, disciplined, and controlled by power devices to make it useful and productive within a given social context. Institutional violence thus manifests as a power device that marginalizes dissident bodies.

For authors such as Rocon et al. [45], the concept of the docile body is evident in the transgender healthcare process within SUS, an important health policy designed to serve the transgender population. According to these authors, in addition to the process being considerably long from its very onset, it presents highly selective eligibility criteria, forcing transgender individuals to conform to diagnostic parameters of either cisgender men or women without accommodating diverse possibilities such as fluid or agender identities. It is thus understood that there are attempts to discipline or normalize [34] transgender bodies within healthcare institutions when social names are not respected, when the legitimacy of the individual and their health demands are questioned, and/or when the pathologization of transgender bodies is maintained.

Therefore, although there are quality public policies in the country, such as the National LGBT Health Policy [46] and the transgender healthcare process, their implementation is constantly challenged by macro- and micropower relations that permeate institutions across various sectors. Without effectively confronting the institutional dynamics that sustain violence against the transgender population, any inclusion policy risks becoming more an instrument of control than of emancipation.

## 11. Consequences of Institutional Violence for the Health of Transgender People in Brazil

On the basis of the analysis of Foucault’s biopolitics [47], the management of life operates through norms that define which bodies are considered “normal” or “deviant” within a society. By denying transgender people full, welcoming, and qualified access to healthcare services, the system reinforces a logic of exclusion that can be understood as a form of biopolitical control. In this context, institutional violence exemplifies how biopolitics functions to produce discourses that legitimize certain identities while marginalizing others, thereby determining which lives receive care and protection and which are relegated to exclusion and neglect.

Rocon et al. [5] reported that, owing to their fear of discrimination in healthcare settings, transgender individuals often avoid initiating or discontinuing essential health treatments, exacerbating a phenomenon termed “exclusion from access.” These authors emphasize that, given their social vulnerability, experiences of violence in the health sector can aggravate critical conditions for transgender survival, contributing to increased mortality rates.

White-Hughto et al. [48] provide a theoretical model demonstrating how stigma operates at multiple levels—individual, interpersonal, and structural—thereby undermining transgender health. According to these authors, stigma restricts access to resources, such as healthcare, and leads to adverse physical and mental health outcomes, complementing Foucauldian analysis by linking power dynamics directly to health inequalities.

With respect to life expectancy, transgender people in Brazil face an extreme reality of social disadvantage. While the general life expectancy in the country is 76.4 years (73.1 for men and 79.7 for women) [49], the transgender population’s life expectancy is estimated at only 30–35 years [50,51]. This alarming statistic is not an isolated figure but reflects a broader context of structural and institutional violence across multiple sectors—including health, education, and housing—that profoundly impacts both quality of life and survival.

The dossier Murders and Violence Against Brazilian Travestis and Transsexuals, produced by the National Association of Travestis and Transsexuals (ANTRA), starkly illustrates this reality. Between 2017 and 2023, 1057 murders of transgender, travesti, and nonbinary individuals were recorded in Brazil [52], placing the country first worldwide in the ranking of transgender murders—a position it has held for 17 consecutive years. These data reveal that the reduced life expectancy of the transgender population is directly linked to lethal violence, social exclusion, and institutional neglect, underscoring the urgent need for concrete actions to address these grave human rights violations both within and beyond healthcare services.

## 12. Resistance Strategies

The exercise of power is not absolute; it is constantly challenged and reconfigured by the actions and resistances of individuals [13]. The presence of transgender people in healthcare services and their demand for humane and respectful care act as a counterpoint to the hegemonic logic that seeks to control, normalize, and pathologize their bodies and identities. By occupying these spaces and asserting their rights, they foster shifts in institutional discourse, challenging the mechanisms of control and exclusion that have historically regulated access to care.

These acts of resistance, although often silent or individual, carry political significance. They create cracks in normative practices that perpetuate marginalization and compel institutions to confront their own limitations and prejudices. By questioning processes such as mandatory pathologizing diagnoses, denial of hormone treatments, or refusal of gender-affirming surgeries, transgender individuals contribute to reshaping the foundations upon which healthcare services operate [40]. These movements, even when seemingly small, have the potential to destabilize historical structures of exclusion and create space for broader institutional and social change [47].

Moreover, these acts of resistance are not limited to the individual level. The voices and demands of collectives, associations, and social movements have increased, advocating for inclusive public policies and institutional transformations that recognize gender diversity [40]. A concrete example is the Transcidadania Program implemented in São Paulo [53], which integrates social, educational, and healthcare initiatives aimed at the social reintegration of transgender individuals. This program demonstrates how institutional mechanisms, when reoriented toward inclusion, can effectively expand access to rights and promote a culture of recognition within public services. Its existence illustrates how popular mobilization, political advocacy, and collective organization can disrupt traditional dynamics of exclusion and generate sustainable change.

Similar processes have been observed in other contexts. Ghosh [54], for example, shows how internal pressure and the politics of alignment led major U.S. companies to adopt inclusive healthcare benefits for transgender employees, demonstrating that institutional transformation becomes possible when social mobilization intersects with political commitment and administrative responsibility.

The resistance of transgender individuals exposes the failure of exclusionary models and encourages collective reflection on the importance of practices that respect the dignity and subjectivity of each person [47]. Consequently, confronting oppressive norms in healthcare services is not limited to the transgender population; it generates a broader social impact, challenges disciplinary power, and opens pathways for more just and inclusive relationships.

## 13. Proposals for Change

Ghosh [43] emphasized that a lack of professional training and systemic biases are central factors driving inequality in healthcare. She argues that strengthening cultural competence through education, policy reforms, and community engagement is essential for reducing health disparities.

Moscheta [55] suggested that healthcare workers must make concerted efforts to respond effectively to the mechanisms of oppression affecting the LGBT population as a whole. The author proposes responsive communication as a strategy to prevent, among other challenges, transphobia in healthcare services. This approach requires healthcare professionals to reject certainties and adopt a curious openness toward the patient, seeking to understand the individual not through the provider’s values but through the legitimization of a non-oppressive relationship.

The bonds formed between healthcare workers and the individuals they serve are decisive in establishing trust, which in turn influences how users engage with public health programs [56].

In this context, it is crucial to develop strategies that promote the social inclusion of marginalized or minority groups, grounded in the principle of equity [57]. Equity entails ensuring that vulnerable populations, such as the transgender population, have equal access to healthcare services and that disparities in health outcomes between different population groups are reduced [11]. This approach recognizes that greater support and resources must be provided to the most disadvantaged.

The concept of affirmative action [58] is another important tool aligned with social justice principles [57]. Affirmative action seeks to redress historical exclusion, creating unequal conditions intentionally designed to benefit groups that have been marginalized [58].

Addressing the challenges highlighted in this study requires a comprehensive approach, ranging from implementing effective public policies and training healthcare professionals to the structural transformation of healthcare systems, all aimed at ensuring welcoming, respectful, and equitable care for transgender individuals.

These propositions align with those of the study by Reisner et al. [59], which highlights that stigma, social exclusion, and institutional failure lead to dramatically worse health outcomes for transgender individuals worldwide. The authors advocate for inclusive and affirmative healthcare systems, emphasizing the adoption of gender affirmation as a public health framework and the improvement of data collection and policies aimed at reducing transgender health inequities. Consequently, the institutional violence and neglect observed in Brazil reflect a broader international pattern, reinforcing the urgency of comprehensive and structural institutional changes.

Therefore, respecting gender identity in social and institutional contexts, particularly within healthcare, is clearly fundamental to promoting equity, dignity, and human rights. Recognizing and valuing a person’s gender identity goes beyond ethical consideration; it is a prerequisite for ensuring that all individuals, regardless of their gender expression or identity, have access to comprehensive, high-quality care free from discrimination. Respect for gender identity directly impacts healthcare access and adherence and contributes to reducing social marginalization.

## 14. Conclusions

Foucauldian theory leads us to understand how power relations sustain exclusionary practices within healthcare services. Linking violence against transgender people with the Microphysics of Power allows us to comprehend that this violence is not only physical or explicit but also structural, symbolic, and relational. Everyday practices within healthcare services, if not properly directed and managed, can perpetuate a logic of control, exclusion, and dehumanization, revealing a diffuse form of power that shapes who may or may not access basic rights such as healthcare. As noted by the authors, institutional violence occurs both through the overt denial of care and through covert mechanisms that regulate access, create barriers, and reaffirm social hierarchies within the very spaces that should promote equity. Unveiling violence within healthcare services is, therefore, essential for a more accurate and profound understanding of the situation. It is from this understanding that quality, comprehensive, and intersectoral care practices can be effectively offered. Finally, given the particularities of this group (transgender individuals), the creation and/or continuation of professional training programs, such as ongoing health education, is encouraged as a means to improve communication and the reception of the transgender population.

## Data Availability

The original contributions presented in this study are included in the article. Further inquiries can be directed to the corresponding author.

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
