# Peer review of "Institutional Violence Perpetrated Against Transgender Individuals in Brazilian Healthcare Services: A Conceptual Analysis Based on Foucault’s Microphysics of Power"

_ijerph, 2025, doi:10.3390/ijerph22111655_

Round 1

Reviewer 1 Report

Comments and Suggestions for Authors

Your essay addresses a critically important topic—the institutional violence faced by transgender individuals in healthcare—through a theoretical lens grounded in Michel Foucault’s Microphysics of Power. The subject matter is highly relevant and timely, and the paper’s strengths include a comprehensive review of multiple forms of violence (symbolic, structural, psychological, physical) and a thoughtful connection to power dynamics in healthcare settings. The discussion clearly conveys the urgency of the problem, supported by recent data (e.g., high rates of care avoidance and dire life expectancy statistics). The essay format allows for a wide-ranging reflective analysis. However, major revisions are required to enhance the manuscript’s academic rigor, clarity, and theoretical integration before it can be considered for publication. Below, I detail the key areas that need improvement:

Introduction and Literature Context

The introduction provides a generally solid background on trans health disparities and institutional violence, but it can be improved to ensure all relevant literature and context are included. The current introduction focuses heavily on Brazilian context and Foucauldian theory, which is appropriate, yet it should better acknowledge the broader international and theoretical landscape. For instance, incorporating more global perspectives on transgender health disparities would strengthen the background. A reference such as Reisner et al. (2016) in The Lancet underscores that transgender people worldwide face stigma, exclusion, and worse health outcomes, highlighting that the issues identified are not isolated to Brazil. Including such sources will position your study within the global public health concern that transgender healthcare represents.

Theoretical Framework

While Foucault’s Microphysics of Power is a valuable lens (and you explain it well), the manuscript would benefit from engaging with additional frameworks from medical sociology and gender studies to deepen the analysis of institutional violence. One notable omission is the literature on stigma and minority stress. The manuscript repeatedly notes that fear of discrimination leads transgender individuals to avoid care—a point well-documented in prior research. Citing a critical review like Hughto et al. (2015) would provide a theoretical model for how stigma operates at multiple levels (individual, interpersonal, structural) to harm trans health.

This work demonstrates that transgender stigma restricts access to resources (like healthcare) and leads to adverse physical and mental health outcomes, complementing your Foucauldian analysis by linking power dynamics directly to health disparities. Incorporating the stigma/minority stress perspective can enrich the theoretical foundation and show readers that your findings align with established social determinants of health frameworks (e.g., Meyer’s minority stress model, which has been extended to transgender populations by Hendricks & Testa, 2012, as noted in your text). Also, an intersectional analysis should be strengthened. The current manuscript touches on vulnerabilities (e.g., social, economic marginalization) but does not explicitly analyze how intersecting factors such as race, class, or sexuality compound the violence trans people face. As Ghosh (2025) emphasizes in a recent review, a lack of attention to intersectionality in transgender healthcare research is a gap that needs addressing. Your discussion of institutional violence would be more nuanced if you acknowledge, for example, that trans women of color or those of low socioeconomic status may experience compounded institutional biases. Even if detailed intersectional analysis is beyond your scope, a brief recognition of these layered inequities (with supporting references) would improve the manuscript’s depth.

Research Design/Methodology

The essay is described as a “theoretical-reflective study” without strict criteria for literature selection. While a flexible, narrative approach is acceptable for a reflective essay, the current description raises concerns about reproducibility and potential bias. It is essential to improve the transparency and rigor of your methodology section (or clearly delineate it within the introduction). Readers (and reviewers) need to understand how you engaged with the literature: How did you decide which sources to include? Was there an attempt to capture a comprehensive range of studies on violence against trans people in healthcare, or was the selection based on convenience/known sources?

The manuscript would benefit from a clearer explanation of your literature search and inclusion strategy, even if it was not a formal systematic review. For example, you might state that you focused on recent (last 5–10 years) scholarly works on transgender health care discrimination, supplemented by foundational theoretical texts (Foucault, Bourdieu, etc.) and relevant Brazilian policy documents. If your own 2024 systematic review (Leal et al., 2024) was used as a starting point, mention how its findings guided this reflective analysis. Clarifying this will assure readers that your arguments rest on a broad and credible literature base and not just a select subset of convenient references. Moreover, consider framing the paper explicitly as a narrative review or conceptual analysis in the title or abstract, so that readers and indexers recognize the nature of the work.

Content Organization and Coherence

The manuscript’s structure generally flows logically through various facets of the problem (definitions of violence, manifestations in healthcare, historical context, challenges, theoretical interpretation, consequences, resistance, and proposals for change). The use of subheadings is effective. To further improve clarity, ensure that each section explicitly ties back to the core theme of institutional violence. For example, in the historical background section, after describing the colonial and dictatorial legacy of transphobia in Brazil, explicitly link that history to present institutional norms (this connection is implied but could be more explicit).

In the sections on “Violence in healthcare” (symbolic, structural, psychological, physical violence), consider adding a few sentences to synthesize how these forms collectively reinforce institutional violence, rather than just listing examples. Currently, these subsections are informative, but the reader would benefit from a concluding remark that ties them together under the umbrella of Foucault’s diffuse power concept—essentially, showing that each form of micro-level violence accumulates into the macro phenomenon of institutionalized exclusion. This synthesis will highlight the originality of your Foucauldian approach: rather than merely cataloguing types of mistreatment, you are illustrating how they produce “docile bodies” and systematic marginalization, aligning with Foucault’s thesis (perhaps referencing Discipline and Punish concepts as you already began to do).

Depth of Analysis

Some parts of the manuscript would benefit from further elaboration or clarification. One such area is the “Resistance strategies” section. You rightly note that the presence and agency of trans individuals in healthcare is itself a form of resistance that can provoke institutional change. This section could be strengthened with more concrete examples or references. For instance, you mention social movements and policies (like the Transcidadania program) that fight for inclusive change. It would reinforce your point to cite evidence of their impact. Consider referencing work on institutional change in other sectors as a parallel, to show that progress is possible: Ghosh (2021) provides a compelling example in the corporate sector, where internal advocacy and policy alignment (via the Corporate Equality Index) led Fortune 500 companies to rapidly adopt transgender-inclusive healthcare benefits. Bringing in this reference can broaden your analysis of resistance by illustrating how engaging institutions with concrete policy demands—the politics of alignment—resulted in a “quiet transgender revolution” in workplace benefits. This analogy underscores that institutions can be pressured to change practices towards trans inclusion—a hopeful counterpoint to the grim status quo in healthcare. Including such insights would enrich your discussion of how power structures might be challenged and transformed.

Use of Evidence and References

Overall, you have grounded your arguments in relevant literature, especially Brazilian studies and some international sources. To further improve, ensure that each major claim is supported by the most pertinent and up-to-date references. There are a few places where additional citations are needed or would add significant weight. For example, when discussing healthcare avoidance and mistrust, cite quantitative findings to reinforce the narrative. You mention the phenomenon termed “exclusion from access” (Rocon et al., 2018) due to fear of discrimination. Complement this with data from large-scale studies: the U.S. Transgender Survey 2015 found that 23% of respondents avoided doctor visits out of fear of mistreatment and 33% had at least one negative experience with a provider in the past year (Johnson et al., 2019). Including such statistics (alongside your existing Brazilian data from Costa et al., 2018 and Silva et al., 2022) will drive home the magnitude of the issue.

Furthermore, when asserting that transgender people face greater healthcare discrimination than other groups, you could reference Kcomt (2019)’s rapid systematic review, which concluded that transgender individuals encounter profoundly higher rates of healthcare discrimination compared to sexual minorities and the general population. This citation would bolster your claims about the unique severity of institutional bias against trans people and emphasize the necessity of your study.

In the “Proposals for change” section, you rightly call for provider training and cultural shift; here, referencing a source on cultural competency training efficacy would be apt. Ghosh (2025) explicitly reviews strategies like education, policy reform, and community collaboration to improve cultural competence in transgender healthcare. Citing this work can lend empirical support to your recommendation that better provider training and inclusive policies will mitigate institutional violence. It can also provide a segue to mention concrete measures (e.g., integrating transgender health content into medical curricula, as other authors have suggested)—showing that your proposals are not just idealistic, but grounded in best practices from the literature.

Lastly, ensure all in-text citations are accurate and appear in the reference list. A quick scan suggests the references are comprehensive and up-to-date (many 2023–2024 studies, which is excellent). Adding the suggested key references below will further improve the manuscript’s scholarly foundation.

Clarity and Language

The manuscript’s English is generally understandable and the academic tone is appropriate, but there are instances of awkward phrasing and minor grammatical issues. For example, some sentences are overly long or have disrupted flow. These should be corrected for clarity. Ensure consistent use of terms (at times “transsexual” is used in historical context vs. “transgender” elsewhere; make sure the usage is context-appropriate). I recommend a thorough copy-editing pass to improve readability—for instance, simplifying complex sentences, checking subject-verb agreement, and eliminating redundant words. Improving the English will not only enhance reader comprehension but also better showcase the quality of your analysis. If writing in English is a challenge, seeking assistance from a native speaker or professional editor would be worthwhile. The goal is to present your important findings in the clearest and most compelling manner possible.

In summary, this manuscript has a valuable focus on a pressing healthcare issue and uses a strong theoretical viewpoint. With major revisions—particularly in clarifying methodology, expanding the theoretical framework, integrating additional key literature, and improving structural and language clarity—the paper can make a meaningful contribution to understanding and ultimately combating the institutional violence that transgender people face. I encourage the authors to take advantage of the rich body of related research (some referenced above) to address the noted gaps. Revisions should aim to transform the paper from a broad essay into a tighter, more analytically rigorous scholarly article. The effort will be well justified by the importance of the topic. I look forward to reviewing a substantially improved version of this work.

Suggested key references to strengthen the manuscript: (the authors should incorporate these to enhance theoretical grounding and context)

  • Ghosh, A. (2025). “Cultural Competence in Transgender Healthcare.” In E. Leung (ed.), Advancing Equity - Health, Rights, and Representation in LGBTQ+ Communities. IntechOpen Publishing.—This very recent book chapter provides an in-depth review of strategies for improving cultural competence in transgender healthcare. It emphasizes that lack of provider training and systemic biases are key drivers of inequitable care, and argues that strengthening cultural competence (through education, policy changes, and community engagement) is vital for reducing health disparities. Citing this work would support your recommendations for professional training and institutional reform, offering evidence-based solutions to the problems you identify.
  • Ghosh, Apoorva. (2021). “The Politics of Alignment and the ‘Quiet Transgender Revolution’ in Fortune 500 Corporations, 2008–2017.” Socio-Economic Review 19(3): 1095–1125.—This sociology article analyzes how internal advocacy and policy mandates led major U.S. corporations to adopt transgender-inclusive health benefits over the past decade. The findings illustrate how institutional norms can shift in response to coordinated pressure using the politics of alignment. Incorporating this reference in your discussion of resistance and change will broaden the context, suggesting that even entrenched institutions like healthcare systems can be moved toward inclusivity through similar alignment of policies and incentives,
  • Johnson, A. H., Hill, I., Beach‑Ferrara, J., Rogers, B. A., & Bradford, A. (2019). “Common barriers to healthcare for transgender people in the U.S. Southeast.” International Journal of Transgender Health 21(1): 70–78.—This peer-reviewed study presents original qualitative data from focus groups with transgender and non‑binary individuals living in the U.S. Southeast, a region that is under‐studied but structurally analogous to marginalized contexts like parts of Brazil. The authors identify four overlapping categories of barriers—fear and mistrust of providers; geographic and insurance-related inconsistency in access; disrespect and misgendering by practitioners; and compound mistreatment due to intersecting race, class, geographic location, and gender identity discrimination—a framework that closely parallels your typology of symbolic, structural, psychological, and physical forms of institutional violence. Including this work strengthens your manuscript by adding empirical support from a comparable context, demonstrating that these multi-level healthcare exclusions occur under different political systems and geographic settings. It further corroborates the interplay between interpersonal provider behaviors and structural policy failures, reinforcing your Foucauldian argument about diffuse power operating at micro and macro levels simultaneously. In short, this citation will anchor your theoretical reflections in real-world qualitative evidence and broaden the manuscript’s comparative relevance.
  • Kcomt, L. (2019). “Profound Health-Care Discrimination Experienced by Transgender People: Rapid Systematic Review.” Social Work in Health Care 58(2): 201–219.—Kcomt’s review quantitatively summarizes the prevalence of healthcare discrimination against transgender individuals in the U.S., comparing it to discrimination against cisgender sexual minorities and the general population. It found that transgender people experience exceptionally high rates of healthcare discrimination and face more barriers to care than other groups, leading to compromised access. This reference would provide strong empirical support for your claims about the pervasiveness of institutional mistreatment. Citing it can help convince readers (and policymakers) that the patterns you discuss are not just anecdotal or localized, but are documented broadly—thereby underlining the significance of addressing institutional violence. It also backs up your point that transphobic discrimination is a distinct problem even relative to homophobia in healthcare, justifying focused attention on the “T” in LGBTQ+ health equity efforts.

  • Reisner, S. L., Poteat, T., Keatley, J., Cabral, M., Mothopeng, T., Dunham, E., ... & Baral, S. D. (2016). Global health burden and needs of transgender populations: a review. The Lancet388(10042), 412-436.—This paper is part of The Lancet’s landmark series on transgender health and provides a global overview of health issues faced by transgender communities. It highlights how stigma, social exclusion, and institutional failures lead to dramatically worse health outcomes for transgender people worldwide, and calls for healthcare systems to become more inclusive and affirming. The authors stress the need for gender-affirmation as a public health framework and improved data collection and policies to address trans health inequities. By referencing this work, you can reinforce that the institutional violence and neglect you describe in Brazil reflect a broader international pattern. It will lend weight to your argument that comprehensive institutional changes (such as those you propose) are urgently needed, and it situates your study as part of a global call to action on transgender health rights.

  • White Hughto, J.M., Reisner, S.L., & Pachankis, J.E. (2015). “Transgender Stigma and Health: A Critical Review of Stigma Determinants, Mechanisms, and Interventions.” Social Science & Medicine 147: 222–231.—This widely-cited review synthesizes literature on how stigma drives health disparities for transgender people. It demonstrates via a social-ecological model that transgender stigma operates at individual, interpersonal, and structural levels to impede access to resources (like healthcare) and harm health. The authors also discuss interventions to mitigate these effects. By citing this work, you would strengthen the theoretical framework of your manuscript, connecting the day-to-day micro-aggressions and institutional policies you describe to a proven model of how stigma translates into poorer health outcomes. It reinforces the notion that institutional violence (as you frame it) is a form of structural stigma, thus bridging your Foucauldian analysis with public health literature on stigma and discrimination.

Integrating these references will not only update your manuscript with cutting-edge research and perspectives, but also help address some of the conceptual gaps noted in this review. They will improve the paper’s theoretical robustness, empirical grounding, and international contextualization, thereby making your arguments more compelling to both academic and policy audiences.

Author Response

Your essay addresses a critically important topic—the institutional violence faced by transgender individuals in healthcare—through a theoretical lens grounded in Michel Foucault’s Microphysics of Power. The subject matter is highly relevant and timely, and the paper’s strengths include a comprehensive review of multiple forms of violence (symbolic, structural, psychological, physical) and a thoughtful connection to power dynamics in healthcare settings. The discussion clearly conveys the urgency of the problem, supported by recent data (e.g., high rates of care avoidance and dire life expectancy statistics). The essay format allows for a wide-ranging reflective analysis. However, major revisions are required to enhance the manuscript’s academic rigor, clarity, and theoretical integration before it can be considered for publication. Below, I detail the key areas that need improvement:

Introduction and Literature Context

The introduction provides a generally solid background on trans health disparities and institutional violence, but it can be improved to ensure all relevant literature and context are included. The current introduction focuses heavily on Brazilian context and Foucauldian theory, which is appropriate, yet it should better acknowledge the broader international and theoretical landscape. For instance, incorporating more global perspectives on transgender health disparities would strengthen the background. 

Response: Dear reviewer, thank you for your comment. We chose to begin the discussion and contextualization by focusing on the Brazilian scenario, considering its political, institutional, and social specificities. We understand that the Unified Health System (SUS) is a unique model, whose dynamics are not fully comparable to other international contexts, which justified the analytical focus adopted. Nevertheless, recognizing the relevance of a broader view, the international context was incorporated throughout the text. It should be noted that, according to the Foucauldian framework that guides the study, the analysis of violence must be situated in specific contexts, since power relations are expressed in a unique way in each social and institutional structure. Thus, we understand that the localized approach does not limit the discussion but, on the contrary, enhances its analytical density and consistency with the theoretical framework adopted.

A reference such as Reisner et al. (2016) in The Lancet underscores that transgender people worldwide face stigma, exclusion, and worse health outcomes, highlighting that the issues identified are not isolated to Brazil. Including such sources will position your study within the global public health concern that transgender healthcare represents.

Response: Dear reviewer, thank you for your comment. The reference has been added to the text. 

Theoretical Framework

While Foucault’s Microphysics of Power is a valuable lens (and you explain it well), the manuscript would benefit from engaging with additional frameworks from medical sociology and gender studies to deepen the analysis of institutional violence. One notable omission is the literature on stigma and minority stress. The manuscript repeatedly notes that fear of discrimination leads transgender individuals to avoid care—a point well-documented in prior research. Citing a critical review like Hughto et al. (2015) would provide a theoretical model for how stigma operates at multiple levels (individual, interpersonal, structural) to harm trans health. This work demonstrates that transgender stigma restricts access to resources (like healthcare) and leads to adverse physical and mental health outcomes, complementing your Foucauldian analysis by linking power dynamics directly to health disparities.

Response: Dear reviewer, thank you for your comment. The reference has been added to the text. 

Incorporating the stigma/minority stress perspective can enrich the theoretical foundation and show readers that your findings align with established social determinants of health frameworks (e.g., Meyer’s minority stress model, which has been extended to transgender populations by Hendricks & Testa, 2012, as noted in your text). Also, an intersectional analysis should be strengthened. The current manuscript touches on vulnerabilities (e.g., social, economic marginalization) but does not explicitly analyze how intersecting factors such as race, class, or sexuality compound the violence trans people face. As Ghosh (2025) emphasizes in a recent review, a lack of attention to intersectionality in transgender healthcare research is a gap that needs addressing. Your discussion of institutional violence would be more nuanced if you acknowledge, for example, that trans women of color or those of low socioeconomic status may experience compounded institutional biases. Even if detailed intersectional analysis is beyond your scope, a brief recognition of these layered inequities (with supporting references) would improve the manuscript’s depth.

Response: Dear reviewer, thank you for your comment. The reference has been added to the text. 

Research Design/Methodology

The essay is described as a “theoretical-reflective study” without strict criteria for literature selection. While a flexible, narrative approach is acceptable for a reflective essay, the current description raises concerns about reproducibility and potential bias. It is essential to improve the transparency and rigor of your methodology section (or clearly delineate it within the introduction). Readers (and reviewers) need to understand how you engaged with the literature: How did you decide which sources to include? Was there an attempt to capture a comprehensive range of studies on violence against trans people in healthcare, or was the selection based on convenience/known sources? The manuscript would benefit from a clearer explanation of your literature search and inclusion strategy, even if it was not a formal systematic review. For example, you might state that you focused on recent (last 5–10 years) scholarly works on transgender health care discrimination, supplemented by foundational theoretical texts (Foucault, Bourdieu, etc.) and relevant Brazilian policy documents. If your own 2024 systematic review (Leal et al., 2024) was used as a starting point, mention how its findings guided this reflective analysis. Clarifying this will assure readers that your arguments rest on a broad and credible literature base and not just a select subset of convenient references. Moreover, consider framing the paper explicitly as a narrative review or conceptual analysis in the title or abstract, so that readers and indexers recognize the nature of the work.

Response: Dear reviewer, thank you for your comment. Following your suggestion, we have included a new section on methodology and sought to improve the approaches adopted. In addition, we have changed the title of the manuscript to “Institutional Violence Against Transgender Individuals in Brazilian Healthcare Services: A Conceptual Analysis Based on Foucault’s Microphysics of Power” 

Content Organization and Coherence

The manuscript’s structure generally flows logically through various facets of the problem (definitions of violence, manifestations in healthcare, historical context, challenges, theoretical interpretation, consequences, resistance, and proposals for change). The use of subheadings is effective. To further improve clarity, ensure that each section explicitly ties back to the core theme of institutional violence. For example, in the historical background section, after describing the colonial and dictatorial legacy of transphobia in Brazil, explicitly link that history to present institutional norms (this connection is implied but could be more explicit).

Response: As suggested, after describing the colonial and dictatorial legacy of transphobia in Brazil, we explain how this history shapes current institutional norms.

In the sections on “Violence in healthcare” (symbolic, structural, psychological, physical violence), consider adding a few sentences to synthesize how these forms collectively reinforce institutional violence, rather than just listing examples. Currently, these subsections are informative, but the reader would benefit from a concluding remark that ties them together under the umbrella of Foucault’s diffuse power concept—essentially, showing that each form of micro-level violence accumulates into the macro phenomenon of institutionalized exclusion. This synthesis will highlight the originality of your Foucauldian approach: rather than merely cataloguing types of mistreatment, you are illustrating how they produce “docile bodies” and systematic marginalization, aligning with Foucault’s thesis (perhaps referencing Discipline and Punish concepts as you already began to do).

Response: As suggested, we summarized the information, seeking to convey the aforementioned connection.

Depth of Analysis

Some parts of the manuscript would benefit from further elaboration or clarification. One such area is the “Resistance strategies” section. You rightly note that the presence and agency of trans individuals in healthcare is itself a form of resistance that can provoke institutional change. This section could be strengthened with more concrete examples or references. For instance, you mention social movements and policies (like the Transcidadania program) that fight for inclusive change. It would reinforce your point to cite evidence of their impact. Consider referencing work on institutional change in other sectors as a parallel, to show that progress is possible: Ghosh (2021) provides a compelling example in the corporate sector, where internal advocacy and policy alignment (via the Corporate Equality Index) led Fortune 500 companies to rapidly adopt transgender-inclusive healthcare benefits. Bringing in this reference can broaden your analysis of resistance by illustrating how engaging institutions with concrete policy demands—the politics of alignment—resulted in a “quiet transgender revolution” in workplace benefits. This analogy underscores that institutions can be pressured to change practices towards trans inclusion—a hopeful counterpoint to the grim status quo in healthcare. Including such insights would enrich your discussion of how power structures might be challenged and transformed.

Response: Dear reviewer, thank you for your comment. The reference has been added to the text. 

Use of Evidence and References

Overall, you have grounded your arguments in relevant literature, especially Brazilian studies and some international sources. To further improve, ensure that each major claim is supported by the most pertinent and up-to-date references. There are a few places where additional citations are needed or would add significant weight. For example, when discussing healthcare avoidance and mistrust, cite quantitative findings to reinforce the narrative. You mention the phenomenon termed “exclusion from access” (Rocon et al., 2018) due to fear of discrimination. Complement this with data from large-scale studies: the U.S. Transgender Survey 2015 found that 23% of respondents avoided doctor visits out of fear of mistreatment and 33% had at least one negative experience with a provider in the past year (Johnson et al., 2019). Including such statistics (alongside your existing Brazilian data from Costa et al., 2018 and Silva et al., 2022) will drive home the magnitude of the issue. Furthermore, when asserting that transgender people face greater healthcare discrimination than other groups, you could reference Kcomt (2019)’s rapid systematic review, which concluded that transgender individuals encounter profoundly higher rates of healthcare discrimination compared to sexual minorities and the general population. This citation would bolster your claims about the unique severity of institutional bias against trans people and emphasize the necessity of your study.

Response: Dear reviewer, thank you for your comment. The reference has been added to the text. 

In the “Proposals for change” section, you rightly call for provider training and cultural shift; here, referencing a source on cultural competency training efficacy would be apt. Ghosh (2025) explicitly reviews strategies like education, policy reform, and community collaboration to improve cultural competence in transgender healthcare. Citing this work can lend empirical support to your recommendation that better provider training and inclusive policies will mitigate institutional violence. It can also provide a segue to mention concrete measures (e.g., integrating transgender health content into medical curricula, as other authors have suggested)—showing that your proposals are not just idealistic, but grounded in best practices from the literature.

Lastly, ensure all in-text citations are accurate and appear in the reference list. A quick scan suggests the references are comprehensive and up-to-date (many 2023–2024 studies, which is excellent). Adding the suggested key references below will further improve the manuscript’s scholarly foundation.

Response: Dear reviewer, thank you for your comment. The reference has been added to the text. 

Clarity and Language

The manuscript’s English is generally understandable and the academic tone is appropriate, but there are instances of awkward phrasing and minor grammatical issues. For example, some sentences are overly long or have disrupted flow. These should be corrected for clarity. 

Response: As suggested, we conducted a linguistic review with the aim of improving the quality of the writing and making it easier to read.

Ensure consistent use of terms (at times “transsexual” is used in historical context vs. “transgender” elsewhere; make sure the usage is context-appropriate). I recommend a thorough copy-editing pass to improve readability—for instance, simplifying complex sentences, checking subject-verb agreement, and eliminating redundant words. Improving the English will not only enhance reader comprehension but also better showcase the quality of your analysis. If writing in English is a challenge, seeking assistance from a native speaker or professional editor would be worthwhile. The goal is to present your important findings in the clearest and most compelling manner possible.

In summary, this manuscript has a valuable focus on a pressing healthcare issue and uses a strong theoretical viewpoint. With major revisions—particularly in clarifying methodology, expanding the theoretical framework, integrating additional key literature, and improving structural and language clarity—the paper can make a meaningful contribution to understanding and ultimately combating the institutional violence that transgender people face. I encourage the authors to take advantage of the rich body of related research (some referenced above) to address the noted gaps. Revisions should aim to transform the paper from a broad essay into a tighter, more analytically rigorous scholarly article. The effort will be well justified by the importance of the topic. I look forward to reviewing a substantially improved version of this work.

Response: Thank you for all your comments. As suggested, we have conducted a thorough review of the study as a whole. We have sought to incorporate all suggestions and remain at your disposal.  

Suggested key references to strengthen the manuscript: (the authors should incorporate these to enhance theoretical grounding and context): 

Ghosh, A. (2025). “Cultural Competence in Transgender Healthcare.” In E. Leung (ed.), Advancing Equity - Health, Rights, and Representation in LGBTQ+ Communities. IntechOpen Publishing.—This very recent book chapter provides an in-depth review of strategies for improving cultural competence in transgender healthcare. It emphasizes that lack of provider training and systemic biases are key drivers of inequitable care, and argues that strengthening cultural competence (through education, policy changes, and community engagement) is vital for reducing health disparities. Citing this work would support your recommendations for professional training and institutional reform, offering evidence-based solutions to the problems you identify

Response: Dear reviewer, thank you for your comment. The reference has been added to the text. 

Ghosh, Apoorva. (2021). “The Politics of Alignment and the ‘Quiet Transgender Revolution’ in Fortune 500 Corporations, 2008–2017.” Socio-Economic Review 19(3): 1095–1125.—This sociology article analyzes how internal advocacy and policy mandates led major U.S. corporations to adopt transgender-inclusive health benefits over the past decade. The findings illustrate how institutional norms can shift in response to coordinated pressure using the politics of alignment. Incorporating this reference in your discussion of resistance and change will broaden the context, suggesting that even entrenched institutions like healthcare systems can be moved toward inclusivity through similar alignment of policies and incentives

Response: Dear reviewer, thank you for your comment. The reference has been added to the text. 

Johnson, A. H., Hill, I., Beach‑Ferrara, J., Rogers, B. A., & Bradford, A. (2019). “Common barriers to healthcare for transgender people in the U.S. Southeast.” International Journal of Transgender Health 21(1): 70–78.—This peer-reviewed study presents original qualitative data from focus groups with transgender and non‑binary individuals living in the U.S. Southeast, a region that is under‐studied but structurally analogous to marginalized contexts like parts of Brazil. The authors identify four overlapping categories of barriers—fear and mistrust of providers; geographic and insurance-related inconsistency in access; disrespect and misgendering by practitioners; and compound mistreatment due to intersecting race, class, geographic location, and gender identity discrimination—a framework that closely parallels your typology of symbolic, structural, psychological, and physical forms of institutional violence. Including this work strengthens your manuscript by adding empirical support from a comparable context, demonstrating that these multi-level healthcare exclusions occur under different political systems and geographic settings. It further corroborates the interplay between interpersonal provider behaviors and structural policy failures, reinforcing your Foucauldian argument about diffuse power operating at micro and macro levels simultaneously. In short, this citation will anchor your theoretical reflections in real-world qualitative evidence and broaden the manuscript’s comparative relevance.

Response: Please take the first comment into consideration. 

Kcomt, L. (2019). “Profound Health-Care Discrimination Experienced by Transgender People: Rapid Systematic Review.” Social Work in Health Care 58(2): 201–219.—Kcomt’s review quantitatively summarizes the prevalence of healthcare discrimination against transgender individuals in the U.S., comparing it to discrimination against cisgender sexual minorities and the general population. It found that transgender people experience exceptionally high rates of healthcare discrimination and face more barriers to care than other groups, leading to compromised access. This reference would provide strong empirical support for your claims about the pervasiveness of institutional mistreatment. Citing it can help convince readers (and policymakers) that the patterns you discuss are not just anecdotal or localized, but are documented broadly—thereby underlining the significance of addressing institutional violence. It also backs up your point that transphobic discrimination is a distinct problem even relative to homophobia in healthcare, justifying focused attention on the “T” in LGBTQ+ health equity efforts.

Response: Dear reviewer, thank you for your comment. The reference has been added to the text. 

Reisner, S. L., Poteat, T., Keatley, J., Cabral, M., Mothopeng, T., Dunham, E., ... & Baral, S. D. (2016). Global health burden and needs of transgender populations: a review. The Lancet388(10042), 412-436.—This paper is part of The Lancet’s landmark series on transgender health and provides a global overview of health issues faced by transgender communities. It highlights how stigma, social exclusion, and institutional failures lead to dramatically worse health outcomes for transgender people worldwide, and calls for healthcare systems to become more inclusive and affirming. The authors stress the need for gender-affirmation as a public health framework and improved data collection and policies to address trans health inequities. By referencing this work, you can reinforce that the institutional violence and neglect you describe in Brazil reflect a broader international pattern. It will lend weight to your argument that comprehensive institutional changes (such as those you propose) are urgently needed, and it situates your study as part of a global call to action on transgender health rights.

Response: Dear reviewer, thank you for your comment. The reference has been added to the text. 

White Hughto, J.M., Reisner, S.L., & Pachankis, J.E. (2015). “Transgender Stigma and Health: A Critical Review of Stigma Determinants, Mechanisms, and Interventions.” Social Science & Medicine 147: 222–231.—This widely-cited review synthesizes literature on how stigma drives health disparities for transgender people. It demonstrates via a social-ecological model that transgender stigma operates at individual, interpersonal, and structural levels to impede access to resources (like healthcare) and harm health. The authors also discuss interventions to mitigate these effects. By citing this work, you would strengthen the theoretical framework of your manuscript, connecting the day-to-day micro-aggressions and institutional policies you describe to a proven model of how stigma translates into poorer health outcomes. It reinforces the notion that institutional violence (as you frame it) is a form of structural stigma, thus bridging your Foucauldian analysis with public health literature on stigma and discrimination.

Response: Dear reviewer, thank you for your comment. The reference has been added to the text. 

Integrating these references will not only update your manuscript with cutting-edge research and perspectives, but also help address some of the conceptual gaps noted in this review. They will improve the paper’s theoretical robustness, empirical grounding, and international contextualization, thereby making your arguments more compelling to both academic and policy audiences.

Reviewer 2 Report

Comments and Suggestions for Authors

The aim of this study was to analyze institutional violence perpetrated against transgender individuals in Brazilian healthcare services through the lens of Foucault’s Microphysics of Power.

However, while the authors made a reasonable attempt to achieve the aim of the article, it was not organised and the study did not offer significant academic value. This work is largely a collection or catalog of reports from various bodies of literature, or summarization – if you like. The work is devoid of synthesis or any integrated narrative about the subject matter of using Foucault’s microphysics of power to analyze institutional violence perpetrated against transgender individuals in Brazilian healthcare services. This explains why I struggled to read any new angle or insight into the topic. This is also the reason the work lacks depth and nuance.

I understand the concept of Foucault’s Microphysics of Power and I will be excited to see how that is applied in discussing the institutional violence against transgender individuals in healthcare services in Brazil or any other place. While this study set out to achieve that, it was not sufficiently integrated into the analysis of the literature review.

While not entirely related, the authors have a lesson or two to learn from Gastaldo’s article (also a literature review). Do not cite. It bears no direct connection with the work under review, except that it is also based on Michel Foucault's work. The essence of including it here is so that the authors can read that work and see the sort of rigour and integrated narrative that I expected.  :

Gastaldo, D. (2024). Transgressive Acts: Michel Foucault's Lessons on Resistance for Nurses. Nursing Philosophy, 26(1), e70008. https://doi.org/10.1111/nup.70008

The work lacks methodological clarity and depth.

For instance, Lines 96-101, the explanation provided for not engaging an adequate and relevant search strategy and data synthesis was not clear. You have to meticulously design a search strategy and anchor this research on it. I strongly recommend that the authors do a systematic literature review. 3 pages (25% of the work) was dedicated to summarising dimensions of violence. The rest of the document lacks any synthesis and relation to the aim of the study.

By saying that they “did not adhere to rigid criteria for the selection of bibliographic material. Instead, the theoretical framework was constructed through successive engagements with the theme in existing bibliographic and scientific literature, as well as through the authors' critical reflections”, the authors admitted that they only merely selected supporting texts while overlooking scholarship that complicates or challenges the argument. Even for an essay, cherry-picking evidence makes a flawed academic exercise.

Lines 51 – 54: I am afraid the authors are risking overgeneralization here. There is no doubt about the expansion of the Unified Health System in Brazil and that there a reasonable level of shift but to declare “that healthcare provision for the trans population in Brazil has shifted from a model focused exclusively on gender-affirming surgeries (such as genital reconstruction and mastectomy) to a more personalized and comprehensive model” potentially misleads readers, because there is no evidence of absolute and comprehensive shift in Brazil, except for few urban areas. I know that, in some locales in Brazil, trans care still remains surgery-focused for many patients.

In some cases, research has shown that “health care for transvestites and transsexuals in Brazil is still exclusionary, fragmented, centered on specialized care and guided by curative actions centered on specialized care and guided by curative actions, resembling the care models that preceded the SUS” (Lim et al., 2023).

Lines 55 – 57: “such practices are influenced by the intentions within the user–health service dyad, as well as by market-driven offers.” Can you also add public policy frameworks, professional training, social movement, etc.? (those are the meso-level of influence to healthcare provision for the trans population)

Ref:

Lima, R. R. T. D., Flor, T. B. M., & Noro, L. R. A. (2023). Systematic review on health care for transvestites and transsexuals in Brazil. Revista de Saúde Pública57, 19.

Author Response

The aim of this study was to analyze institutional violence perpetrated against transgender individuals in Brazilian healthcare services through the lens of Foucault’s Microphysics of Power.

However, while the authors made a reasonable attempt to achieve the aim of the article, it was not organised and the study did not offer significant academic value. This work is largely a collection or catalog of reports from various bodies of literature, or summarization – if you like. The work is devoid of synthesis or any integrated narrative about the subject matter of using Foucault’s microphysics of power to analyze institutional violence perpetrated against transgender individuals in Brazilian healthcare services. This explains why I struggled to read any new angle or insight into the topic. This is also the reason the work lacks depth and nuance.

Response: Dear reviewer, thank you for your comment. We try to articulate by providing summaries in each block.

I understand the concept of Foucault’s Microphysics of Power and I will be excited to see how that is applied in discussing the institutional violence against transgender individuals in healthcare services in Brazil or any other place. While this study set out to achieve that, it was not sufficiently integrated into the analysis of the literature review.

Response: Dear reviewer, thank you for your comment. We sought to broaden the discussion in the text as a whole. We hope that the new version brings greater clarity and depth to the text. 

While not entirely related, the authors have a lesson or two to learn from Gastaldo’s article (also a literature review). Do not cite. It bears no direct connection with the work under review, except that it is also based on Michel Foucault's work. The essence of including it here is so that the authors can read that work and see the sort of rigour and integrated narrative that I expected.  :

Gastaldo, D. (2024). Transgressive Acts: Michel Foucault's Lessons on Resistance for Nurses. Nursing Philosophy, 26(1), e70008. https://doi.org/10.1111/nup.70008

Response: Dear reviewer, thank you for your comment. Ee have included a new section on methodology and sought to improve the approaches adopted. 

The work lacks methodological clarity and depth.

For instance, Lines 96-101, the explanation provided for not engaging an adequate and relevant search strategy and data synthesis was not clear. You have to meticulously design a search strategy and anchor this research on it. I strongly recommend that the authors do a systematic literature review. 3 pages (25% of the work) was dedicated to summarising dimensions of violence. The rest of the document lacks any synthesis and relation to the aim of the study.

Response: This study presents a conceptual analysis based on the previously published systematic review article entitled “Institutional Violence Perpetrated against Transgender Individuals in Health Services: A Systematic Review of Qualitative Studies” (Leal et al., 2024). doi: https://doi.org/10.3390/ijerph21081106. 

By saying that they “did not adhere to rigid criteria for the selection of bibliographic material. Instead, the theoretical framework was constructed through successive engagements with the theme in existing bibliographic and scientific literature, as well as through the authors' critical reflections”, the authors admitted that they only merely selected supporting texts while overlooking scholarship that complicates or challenges the argument. Even for an essay, cherry-picking evidence makes a flawed academic exercise.

Response: This study presents a conceptual analysis based on the previously published systematic review article entitled “Institutional Violence Perpetrated against Transgender Individuals in Health Services: A Systematic Review of Qualitative Studies” (Leal et al., 2024). doi: https://doi.org/10.3390/ijerph21081106. 

Lines 51 – 54: I am afraid the authors are risking overgeneralization here. There is no doubt about the expansion of the Unified Health System in Brazil and that there a reasonable level of shift but to declare “that healthcare provision for the trans population in Brazil has shifted from a model focused exclusively on gender-affirming surgeries (such as genital reconstruction and mastectomy) to a more personalized and comprehensive model” potentially misleads readers, because there is no evidence of absolute and comprehensive shift in Brazil, except for few urban areas. I know that, in some locales in Brazil, trans care still remains surgery-focused for many patients.

In some cases, research has shown that “health care for transvestites and transsexuals in Brazil is still exclusionary, fragmented, centered on specialized care and guided by curative actions centered on specialized care and guided by curative actions, resembling the care models that preceded the SUS” (Lim et al., 2023).

Response: Thanks for your comments. We rewrote the paragraphs and included the suggested reference.  

“These global patterns of heteronormativity are also observable in Brazil, where the healthcare system mirrors both the advances and the persistent limitations in meeting the needs of trans people [6,7]. Costa et al. [6] noted that healthcare for the trans population in Brazil has been transitioning from a model focused exclusively on gender-affirming surgeries (such as transgenitalization and mastectomy) to one that is comprehensive and person-centered, particularly in major urban centers. However, Lima et al. [7] warned that healthcare for trans individuals in Brazil remains exclusionary, fragmented, and overly specialized, guided by curative actions and resembling care models that predate the establishment of the Unified Health System (Sistema Único de Sáude - SUS). The purpose of this study is not to conduct an in-depth analysis of this issue, as such practices depend on the intentionality between users and healthcare services, as well as on market-driven dynamics. They also rely on other intermediate-level factors, such as public policy frameworks, professional training, and the influence of social movements, all of which shape the provision of healthcare for the trans population [7].”

Lines 55 – 57: “such practices are influenced by the intentions within the user–health service dyad, as well as by market-driven offers.” Can you also add public policy frameworks, professional training, social movement, etc.? (those are the meso-level of influence to healthcare provision for the trans population)

Response: Thank you for your comment. We have included the suggested excerpts. We have sought to incorporate all suggestions and remain at your disposal.   

Round 2

Reviewer 1 Report

Comments and Suggestions for Authors

Please correct the sentence (p. 10/14, para 4) to:

"Ghosh [54], for example, shows how internal pressure and the politics of alignment led major U.S. companies to adopt inclusive healthcare benefits for transgender employees,..."

Author Response

Dear reviewer,
Thank you for your comment. We have made the requested adjustments. 

Reviewer 2 Report

Comments and Suggestions for Authors

The authors have effected all the major corrections suggested in my previous review. 

Author Response

Dear reviewer,
Thank you for your comment.